# Surfactin and Spo0A-Dependent Antagonism by *Bacillus subtilis* Strain UD1022 against *Medicago sativa* Phytopathogens

**DOI:** 10.3390/plants12051007

**Published:** 2023-02-23

**Authors:** Amanda Rosier, Maude Pomerleau, Pascale B. Beauregard, Deborah A. Samac, Harsh P. Bais

**Affiliations:** 1Department of Plant and Soil Sciences, University of Delaware, 311 AP Biopharma, 590 Avenue 1743, Newark, DE 19713, USA; 2Département de Biologie, Bureau D8-1014, Université de Sherbrooke, 2500 boul. Université Sherbrooke, Sherbrooke, QC J1K 2R1, Canada; 3USDA-ARS Plant Science Research Unit, 1991 Upper Buford Circle, St. Paul, MN 55108, USA

**Keywords:** PGPR, alfalfa, antagonism, *Bacillus subtilis*, *Phytophthora medicaginis*, *Ascochyta medicaginicola*, *Phoma medicaginis*, surfactin, biofilm

## Abstract

Plant growth-promoting rhizobacteria (PGPR) such as the root colonizers *Bacillus* spp. may be ideal alternatives to chemical crop treatments. This work sought to extend the application of the broadly active PGPR UD1022 to *Medicago sativa* (alfalfa). Alfalfa is susceptible to many phytopathogens resulting in losses of crop yield and nutrient value. UD1022 was cocultured with four alfalfa pathogen strains to test antagonism. We found UD1022 to be directly antagonistic toward *Collectotrichum trifolii*, *Ascochyta medicaginicola* (formerly *Phoma medicaginis*), and *Phytophthora medicaginis*, and not toward *Fusarium oxysporum* f. sp. *medicaginis*. Using mutant UD1022 strains lacking genes in the nonribosomal peptide (NRP) and biofilm pathways, we tested antagonism against *A. medicaginicola* StC 306-5 and *P. medicaginis* A2A1. The NRP surfactin may have a role in the antagonism toward the ascomycete StC 306-5. Antagonism toward A2A1 may be influenced by *B. subtilis* biofilm pathway components. The *B. subtilis* central regulator of both surfactin and biofilm pathways Spo0A was required for the antagonism of both phytopathogens. The results of this study indicate that the PGPR UD1022 would be a good candidate for further investigations into its antagonistic activities against *C. trifolii*, *A. medicaginicola*, and *P. medicaginis* in plant and field studies.

## 1. Introduction

The perennial forage legume alfalfa (*Medicago sativa*) is an important crop grown globally and serves as a critical source of animal feed, as well as contributing significantly to nitrogen replenishment of the soil in crop rotations [1]. Soil-resident pathogens such as fungi and oomycetes can significantly impact alfalfa germination, establishment, biomass, and nutritional value of this economically important field crop [2]. Alfalfa yield losses due to foliar pathogens such as spring black stem and leaf spot caused by *Ascochyta medicaginicola* were conservatively averaged to be 13.2% in a study of over 48 harvests across four U.S. states over four years [3]. Great progress has been made in the breeding of alfalfa cultivars resistant to common pathogens [4], however, the challenge remains to develop cultivars optimally resistant to multiple different established and novel pathogens [5,6]. Implementation of alternative strategies such as the use of plant-associated microorganisms and their natural products (‘biologicals’) is a viable tool for the management of these pathogens in conjunction with more traditional methods [7].

A subset of soil and plant root-associated bacteria have long been recognized as having plant beneficial activities including improving plant growth and disease suppression and are known as plant growth promoting rhizobacteria, or ‘PGPR’ [8]. The activities of PGPRs have been intensely researched for use against a myriad of common and devastating soil-borne pathogens. *Bacillus* spp. specifically have been studied for several decades due to their multifaceted modes of plant-beneficial activities [9,10], one of the most robust of which are the direct inhibitory or antagonistic interactions with bacterial and fungal plant pathogens [11]. *Bacillus cereus* (UW85) was found to be significantly suppressive toward *Phytophthora megasperma* f. sp. *medicaginis* in alfalfa tube bioassays, reducing seedling mortality to 0% [12].

Many of the powerful metabolites and antibiotics produced by *Bacillus* spp. have been found to directly suppress microbial pathogens [13]. The largest class of *Bacillus* spp. antibiotics includes nonribosomal peptides (NRP) [14]. The antifungal properties of NRPs are well documented; with their amphiphilic structure, NRPs such as surfactin, iturin, and fengycin/plipastatin can disrupt cell membranes and cause cell death [15].

The *srfAC* operon encodes four different enzyme subunits of the nonribosomal peptide synthase (NRPS) surfactin synthetase complex [16]. NRPs, including plipastatin, require the *sfp* gene product, 4′-phosphopantetheinyl transferase (Sfp), for assembly by the machinery of these complexes [17]. The gene *ppsB* encodes one of five subunits of the NRPS which synthesizes the cyclic NRP plipastatin [18].

The gene *spo0A* encodes a master transcriptional regulator for biofilm [19] and surfactin [20]. SinI derepresses SinR repression of *epsA-O* and *tasA*, triggering matrix production in conjunction with Spo0A [21]. Eps, part of the *epsA-O* operon, encodes for biofilm exopolysaccharide production and the *tasA* gene, part of the *tapA-sipW-tasA* operon, encodes protein fiber components of biofilms; the double mutant *eps*^−^-*tasA*^−^ is completely defective in biofilm formation [22,23].

Another important PGPR characteristic of *Bacillus* spp. is their ability to produce polysaccharide-rich biofilms. *Bacillus* spp. biofilm formation is a hallmark of their multicellular lifestyle, allowing them to adhere to surfaces and respond to environmental conditions through a heterogeneity of cell types within the matrices produced [24]. The ability to form robust biofilms by *Bacillus* spp. is necessary for close association with the plant root [25,26]. *Bacillus* spp. biofilm and NRP synthetic pathways overlap through the activities of various common global regulators [27,28]. On plant roots, the upregulation of biofilm formation appears to be induced by plant root-exuded polysaccharides [29] and surfactin is not required for this biofilm production [30]. However, possible mechanisms and/or specific coordination of genetic pathways have not yet been disentangled for the role of biofilms in *Bacillus* spp. antagonistic activities. 

The growth promotional and plant protective activities of the University of Delaware patented PGPR *B. subtilis* UD1022 are well established for a variety of horticulturally and agriculturally relevant plants [31]. The complete genome of UD1022 has been described [32] and the ability of this strain to produce secondary metabolites, antibiotics, and plant-associated biofilms contributes to its PGPR function [25,33]. The aim of this study was to evaluate whether the PGPR *B. subtilis* UD1022 is antagonistic toward common alfalfa fungal and oomycete pathogens and to investigate possible modes of action for observed antagonisms. Here, we demonstrate the strong antagonistic activity of UD1022 against several alfalfa phytopathogens and compare the likely differences in how these antagonisms are achieved by the PGPR.

## 2. Results

UD1022 antagonism in direct challenge assays ranged from strongly antagonistic to no antagonism toward the different genera of phytopathogens (Figure 1). Indirect culture with UD1022 did not inhibit any of the pathogens in split plate assays (data not shown). *C. trifolii* AN1 and SM were moderately antagonized (40% reduction in colony area) by UD1022 (Figure 1A,B). 

UD1022 had no antagonism toward either strain of *F. oxysporum* f. sp. *medicaginis* (FOM 255 and Fo 3). The greatest antagonism was toward *A. medicaginicola* and *P. medicaginis*. UD1022 antagonism was significant toward both *A. medicaginicola* strains, 63% toward StC 306-5 and 55% towards StC 306-10 (Figure 1E,F). The strain StC 306-5 was selected for further analysis of UD1022 antagonistic mechanisms through the use of select UD1022 functional deletion mutants in NRP and biofilm pathways. UD1022 antagonism against *P. medicaginis* A2A1 was the strongest, reducing colony area by 67% (Figure 1G) and A2A1 was subjected to further analysis to identify a mechanism for the observed activity.

UD1022 NRP mutant srfAC^−^ had a moderate but significantly less antagonistic effect (35% inhibition) on *A*. *medicaginicola* StC 306-5 colony area than UD1022, at nearly 48% inhibition (*p* = 0.044). Colony areas of StC 306-5 responding to ppsB^−^ had 44% inhibition and the double-mutant *ppsB*^−^-*srfAC*^−^ treatment was intermediately inhibited at 39%. Both were statistically similar to the UD1022 treatment (*p* = 0.853 and 0.265, respectively) and to srfAC^−^ (*p* = 0.27 and 0.859, respectively). The fungal colony area response to mutant *sfp*^−^ was not different from the fungal colony in response to UD1022 with 53 % inhibition (*p* = 0.766) (Figure 2). The direct challenge with surfactin at concentrations of 25 and 100 µg/mL resulted in observable but not significant antagonism of StC 306-5 (Appendix A). 

Colony areas of StC 306-5 challenged by UD1022 biofilm mutants *sinI*^−^ or *eps*^−^-*tasA*^−^ were no different from the UD1022 control treatment, remaining antagonistic. Antagonism of the StC 306-5 colony area was reduced to 5% when challenged by the *spo0A*^−^ mutant and the colony area was not statistically different from the water control treatment (*p* = 0.881) (Figure 3). 

For the oomycete, *P. medicaginis* A2A1, the NRP mutants *ppsB*^−^-*srfAC*^−^, *srfAC*^−^, and *sfp*^−^ caused less inhibition (52%, 51%, and 52%, respectively) than UD1022 (62%) and the A2A1 colony areas were significantly different from those in the UD1022 treatment (*p* = 0.002, 0.006, and 0.006, respectively). The colony area of A2A1 challenged with the mutant of *ppsB*^−^ was not significantly different from UD1022 (55% inhibition, *p* = 0.073) (Figure 4). The NRP plipastatin does not seem to contribute to the antagonism of *P. medicaginis* A2A1 to any extent, however, *srfAC*^−^, the double mutant *ppsB*^−^-*srfAC*^−^, and *sfp*^−^ resulted in the mild alleviation of the antagonism (Figure 4). However, the direct application of 100 µg/mL (2 µg total) surfactin had no effect on the colony area (Appendix A). Surfactin and other possible *sfp*-dependent NRPs may have some partial activity in the observed inhibition of the oomycete, however, surfactin alone does not seem to contribute to the activity of UD1022 toward *P. medicaginis* A2A1. 

*The P. medicaginis* A2A1 colony area was statistically less antagonized in treatments of UD1022 biofilm mutants sinI^−^ and *eps*^−^-*tasA*^−^ (51% and 43% inhibition) compared with UD1022 inhibition of 62% (*p* = 0.011 and 0.0002, respectively), and antagonism was reduced to only 24% in treatment with the *spo0A*^−^ mutant (*p* < 0.0001) (Figure 5). The antagonism is strongly *spo0A* dependent and there appears to be a significant influence of the *eps*, *tasA*, and *sinI* genes in the antagonism of UD1022 against A2A1.

## 3. Discussion

Antibiosis of root microflora has long been implicated as one of the many modes of PGPR activity [34]. UD1022 is a PGPR that produces the NRP surfactin (unpublished data) which was proposed to directly inhibit the pathogen *Pseudomonas syringae* pv. *tomato* DC3000 [25]. Further, UD1022 was also shown to have extensive biofilm production and adhesion to plant roots [25,31]. The work presented here demonstrates UD1022’s direct inhibition of common alfalfa pathogens, likely through a mixture of NRP and biofilm interactions.

*Collectotrichum* spp. are the causal agent of anthracnose and *C. trifolii* causes damage to plant stems and petioles leading to significant yield losses in alfalfa [4]. UD1022 antagonism of 40% toward both *C. trifolii* was moderate compared to its effect on *A. medicaginicola* and *P. medicaginis*. Using *B. amyloliquefaciens* LYZ69, Hu et al. [35] observed inhibition of *Collectotrichum truncatum* mycelial growth by 84.3%. LYZ69 crude extract also inhibited mycelial growth and LC-MS analysis revealed the presence of the lipopeptides fengycin (plipastatin) and bacillomycin D. UD1022 has been shown to produce lipopeptides such as surfactin (unpublished data) and UD1022 antagonism of *C. trifolii* could be due to these antimicrobial metabolites.

*Fusarium* spp. are widespread plant pathogens causing root rot and vascular wilt diseases, especially in young plants, and also quickly infect wounded plants. The complete lack of antagonism toward both strains of *F. oxysporum* f. sp. *medicaginis* was notable (Figure 1C,D). A variety of *Bacillus* spp. Have been shown to inhibit a range of different *Fusarium* spp., including *B. subtilis* subsp. *Spizizenii* MB29, which suppresses the alfalfa pathogen *Fusarium semitectum* [36]. *Bacillus* spp. antagonism seems to have specificity in relation to both the *Bacillus* strain and the species and host specialty of the *Fusarium* [37]. 

The ascomycete *A. medicaginicola* is the causal agent of alfalfa spring black stem and leaf spot [38]. Leaf spot disease significantly impacts alfalfa production through defoliation and reduction of dry matter yields, resulting in reduced protein content and poor forage quality [39]. The strong antagonism of UD1022 toward *A. medicaginicola*, especially strain StC 306-5, motivated further investigation of potential mechanisms.

The oomycete *P. medicaginis* causes root rot in *Medicago* spp. and other legume crops such as chickpea and soybean [40,41]. This phytopathogen is most destructive to the establishment of seedlings and can severely reduce yield [42]. UD1022 antagonism was the greatest against *P. medicaginis* A2A1 with a 67% reduction in colony area; a promising result that prompted further investigation into the mechanisms of activity. 

Many *Bacillus* spp. are reported to carry outantifungal activity through the production of NRPs [43]. The genome of UD1022 contains many antimicrobial/antibiotic genes [32], some of which increase production in the presence of soil bacteria (unpublished data). Taken altogether, the results of UD1022 NRP and biofilm mutant antagonism assays with *A. medicaginicola* StC 306-5 suggest that further investigation of UD1022 secondary metabolites is needed, especially sfp-independent products. Considering that the *sinI*^−^ and *eps*^−^-*tasA*^−^ mutants have no significant differences in antagonism, spo0A is likely crucial through pathways unrelated to biofilm production.

The significant alleviation of antagonism by the *spo0A*^−^ mutant is especially interesting. In addition to its role in sporulation and biofilm formation, *spo0A* is also known to be required for surfactin biosynthesis [20] and has been implicated as necessary to produce other *Bacillus* spp. antibiotics including *sfp*-dependent and ribosomally synthesized peptides [20]. Supporting the association between *spo0A* and surfactin, several recent studies have documented the loss of surfactin production in *spo0A*^−^ mutants. Sun et al. [44] found that the surfactin yield in a *spo0A*^−^ mutant of *B. amyloliquefaciens* fmbJ was reduced to ~1.7 mg/liter from that of a wild-type strain of ~5.8 mg/liter. In industrial fermentation experiments, Wang et al. [45] found complete inhibition of surfactin production in an enhanced surfactin construct (*B. subtilis* TS1726 having a strong Pg3 promoter) when *spo0A* was knocked out. Though neither group used the *spo0A* mutants in antagonistic challenge assays, Shao et al. [46] demonstrated relief of antagonism between two strains of *B. velezensis* if *spo0A* was mutated in strain SQR9, resulting in improved biofilm formation and plant benefits by the cocultured wild-type strain FZB42.

UD1022 *sfp*^−^ and *srfAC*^−^ may have some role in the antagonism toward A2A1 (Figure 4), though direct surfactin application showed no antagonism toward the pathogen (Appendix A) suggesting a role for other *sfp*-dependant antibiotics. The *sfp*-dependent NRP iturin A was reported as the mechanism of *B. subtilis* WL-2 biocontrol activity against *P. infestans*, directly inhibiting mycelial growth by 84.9%. Iturin A damaged *P. infestans* mycelia by disrupting cell structure and inducing oxidative stress [47]. Interestingly, this study also found that surfactin had no direct inhibitory effect on mycelia growth, corresponding with the observations with *P. medicaginis* in the current study.

The key genes for biofilm formation found to be associated with the antagonism of UD1022 toward A2A1 could be acting through motility and colonization activities. The ability to spread and move outward or even toward the A2A1 colony may be crucial for full UD1022 antagonistic activity. Colony characteristics of the biofilm mutants, especially *sinI*^−^ and *eps*^−^-*tasA*^−^ cocultured with A2A1 (Figure 5), are strikingly more contained than the amorphous appearance of the wildtype strain UD1022. 

The components of the biofilm pathway are known to be involved in the control of sliding and form of *Bacillus* spp. motility [48]. Sliding motility is a flagellum-independent form of *Bacillus* spp. motility which advances colonies through a synergistic interaction between biofilm-producing and surfactin-producing cells within a population. Gestel et al. [49] found that *eps* and *srfA* are required for effective sliding, proposing that the role of surfactin is to reduce surface friction and the production of exopolysaccharides facilitates the physical alignment of cell bundles along the growing colony front. Considering the role of surfactin in conjunction with the polysaccharide components of biofilms, a double mutant of *eps* and *srfA* would be of interest in future assays.

To elucidate the regulatory components involved in sliding motility, Grau et al. (2015) utilized the ‘sliding only’ *B. subtilis* strain RG4365 and the ‘sliding and swarming proficient’ *B. subtilis* strain NCIB3610 in a series of expression and motility assays [50]. Like Gestel et al. [49], they found that the *eps* gene was required for sliding and, moreover, they showed that sliding motility is induced by low levels of phosphorylated Spo0A, likely through the kinase KinB response to high intracellular potassium concentrations. Mutation of the UD1022 biofilm pathway genes *spo0A* and *eps*, crucial for nonflagellar sliding motility, resulted in significantly less antagonism toward A2A1. Thus, it is possible that UD1022 antagonism toward A2A1 could be reliant on the ability of the cells to be actively engaged in biofilm production and/or sliding.

## 4. Materials and Methods

### 4.1. Strains and Culture Conditions

Four species of alfalfa pathogens were used to test the disease-suppressive activities of UD1022: the ascomycetes *Collectotrichum trifolii* (causing anthracnose) strains AN1 and SM, *Fusarium oxysporum* f. sp. *medicaginis* (causing Fusarium wilt) strains FOM 255 and Fo 3, *Ascochyta medicaginicola* (synonym *Phoma medicaginis* causing spring black stem and leaf spot) strains StC 306-5 and StC 306-10, and the oomycete *Phytophthora medicaginis* (causing Phytophthora root rot) strain A2A1. Strain A2A1 was cultured from plugs on V8 agar, the other strains were cultured on potato dextrose agar (PDA), and all were grown in dark conditions at room temperature (RT). All strains were provided by Deborah Samac of the USDA-ARS, St. Paul, MN, USA.

*B. subtilis* strain UD1022 was cultured from glycerol stocks onto Luria Bertani (LB) agar plates. Cultures were grown overnight at 37 °C. Fresh colonies were selected and grown in 5 mL LB broth shaking at 37 °C to an OD_600_ of ~1.0.

### 4.2. Antagonistic Plate Assays

The suppressive activity of UD1022 against the pathogen strains was evaluated by the mycelial growth rate method [51] with minor modifications. In sterile conditions, 7 mm diameter plugs were taken from the fresh mycelial growth at the edge of the pathogen colony and placed in the center of standard 100 mm petri plates of the appropriate growth medium. Twenty microliter drops of UD1022 fresh culture (OD_600_ ~ 1.0) were pipetted 2 cm from the pathogen plug edge. Sterile water was used to control against the pathogen. 

The antagonistic activity of surfactin against *P. medicaginis* A2A1 and *A. medicaginicola* StC 306-5 was evaluated similarly. Surfactin from *B. subtilis* (Sigma Aldrich, St. Louis, MI, USA) was resuspended in ethanol (10 mg/mL) and diluted in ethanol to 100, 75, 50, and 25 µg/mL. Concentrations were selected to be biologically relevant [52]. Twenty microliters of each dilution were pipetted to 5 mm sterile Whatman paper disks and allowed to dry (equivalent to a total of 2, 1.5, 1, and 0.5 µg surfactin/disk). Disks, including an ethanol control, were placed 2 cm from the edge of the plug on V8 or PDA plates and incubated in the dark at RT for 7 days. Three replicates of all treatments were included per experiment. Each experiment was repeated a total of three times and each experiment was evaluated independently. Uniform photographs were taken of plates daily for 7 days. Data are shown for day seven. The inhibition rate of the pathogen by UD1022 or surfactin was calculated with the formula:*I* (%) = [(*C* − *T*)/(*C* − 0.7)] × 100
where *I* is the inhibition rate, *C* is the area (cm^2^) of the colony in the control, and *T* is the colony area (cm^2^) of the UD1022 treatment. Colony areas for the calculation were measured using ImageJ software [53]. ‘Indirect’ UD1022 antagonistic activity due to bacterial volatiles (BVCs) was tested similarly, except the plug and UD1022 were placed into separate compartments of split petri plates. 

### 4.3. UD1022 Mutant Assays

Based on the initial results, two pathogens were selected for further analysis to interrogate mechanisms of UD1022 antagonism: the ascomycete *A. medicaginicola* StC 306-5 (Figure 1E) and the oomycete *P. medicaginis* A2A1 (Figure 1G). Mutant strains of UD1022 were utilized to interrogate whether the major characteristics shared by *B. subtilis*. are responsible for the antagonisms observed. Mutant strains of UD1022 with deletions in the key nonribosomal peptide (NRP) pathway genes *srfAC*^−^, *sfp*^−^, *ppsB*^−^, double mutant *ppsB*^−^-*srfA*^−^, biofilm genes *spo0A*^−^, *sinI* ^−^, and a double mutant in *eps*^−^ and *tasA*^−^ were used in direct antagonism assays against the two pathogen strains selected. 

Mutations were first introduced in a *B. subtilis* strain (either NCIB 3610, 168, or PY79), and were then introduced in UD1022 using SPP1 phage transduction. [54]. All the mutants except *ppsB*^−^ were an allelic replacement of the orf by the cassette encoding the antibiotic-resistant gene. For *ppsB*^−^, the mutation is a Tn10 transposon insertion (Appendix A). All mutations resulted in functional impairment of the gene through sequence disruption. Mutants were constructed and generously provided by Dr. Pascale Beauregard’s lab (Universite de Sherbrooke, Sherbrooke, QC, Canada). Mutant strains were grown in LB with the appropriate antibiotics (Appendix A). Bacterial cultures were grown to OD_600_ ~1.5 to 2.5, washed once in sterile water, and resuspended to OD_600_ ~1.0 prior to use in the plate antagonism assay as described above. 

### 4.4. Statistical Analysis

Statistical analyses were performed using JMP^®^, Version 16. SAS Institute Inc., Cary, NC, USA, 1989–2021. Data normality (Goodness of Fit using the Shapiro–Wilk test) and homogeneity (Levene test) were reviewed on the residuals prior to analysis of variance (ANOVA). No data transformations were required. Tukey’s HSD One Way Analysis of Variance by treatment was performed for statistical analysis and the *p*-values stated are from the ordered difference report. Statistics were performed on the three treatment replicates (*n* = 3) within each of the three separate experiment repetitions and the results of the most representative experiment are presented.

## 5. Conclusions

UD1022 effectively suppresses the hyphal growth of both the ascomycete *A. medicaginicola* StC 306-5 and the oomycete *P. medicaginis* A2A1 through direct antagonism. UD1022 activity toward StC 306-5 is not attributable to specific components of *Bacillus* spp. biofilm formation. The NRP surfactin appears to contribute to the antagonism, however, this is clearly not the primary cause. Surfactin likely had a greater role in the antagonism of the ascomycete *A. medicaginicola* than in the oomycete *P. medicaginis*. UD1022 biofilm genes *sinI*, *eps*, and *tasA* seem to have a role in the antagonism of the oomycete A2A1, possibly through biofilm motility/matrix activities.

There are several possibilities as to the elements responsible for UD1022 antagonism toward A2A1. There could be alternative *sfp*-dependent lipopeptide or other peptide antibiotics contributing, or the antagonism could be due to some element of the *Bacillus* spp. biofilm-matrix pathways, either through structural or motility interactions, such as *spo0A*-dependent sliding motility.

Interestingly, but somewhat unsurprisingly, *spo0A* is required for UD1022 antagonism against both alfalfa pathogens evaluated. In addition to antibiotic and biofilm activities, Spo0A also regulates *B. subtilis* sporulation [55]. Some research on the effectiveness of sporulating *Bacillus* spp. cultures indicates that there is an increase in antibiotic production and increased biocontrol activity during sporulation [12,56]. Sun et al. [44] found a significant decrease in the production of bacillomycin D, fengycin, and surfactin in *spo0A* knockout mutants of *B. amyloliquefaciens* fmbJ. 

Beyond NRP production, *Bacillus* spp. has an extensive range of identified antimicrobial factors, including ribosomally synthesized peptides and enzymes including chitinases [57] and proteases [58]. Relevant to this work, Spo0A is identified as having a role in regulating some of these factors such as exoprotease production [59], the antibiotic-like Skf peptides [20], and the protease subtilisin [60], which could also be contributing to the overall antagonism activity of UD1022 toward the alfalfa phytopathogens.

These results contribute to the body of literature concluding that *Bacillus* spp. have a multitude of direct antagonistic mechanisms against phytopathogens. Different species of plant-beneficial bacteria have differential modalities toward different genera, species, and biovars of phytopathogens [43,61]. When evaluating *Bacillus* spp. for use as a direct biological or biosynthesis-based product based on their secondary metabolite profile, it is essential to understand that no single metabolite or mechanism will likely work as a broad-spectrum pathogen control agent. Future investigations which interrogate *B. subtilis* UD1022 mechanisms of antagonism molecular, biochemical, and in planta activity will be essential in furthering the applications of PGPRs to crops such as alfalfa.

## Figures and Tables

**Figure 1 plants-12-01007-f001:**
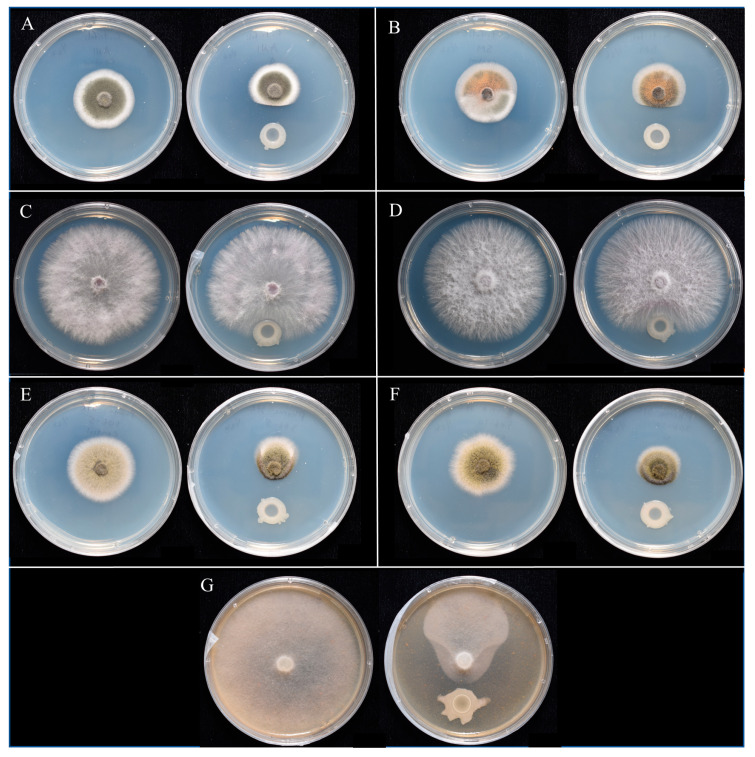
Results of UD1022 direct antagonism assays. Control (left) and treatment plate (right) pairs for each pathogen challenged. (**A**), *Colletotrichum trifolii* AN1. (**B**), *Colletotrichum trifolii* SM. (**C**), *F. oxysporum* f. sp. *medicaginis* Fom3. (**D**), *F. oxysporum* f. sp. *medicaginis* Fom255. (**E**), *Ascochyta medicaginicola* StC 306-5. *(***F**), *Ascochyta medicaginicola* StC 306-10. (**G**), *Phytophthora medicaginis* A2A1.

**Figure 2 plants-12-01007-f002:**
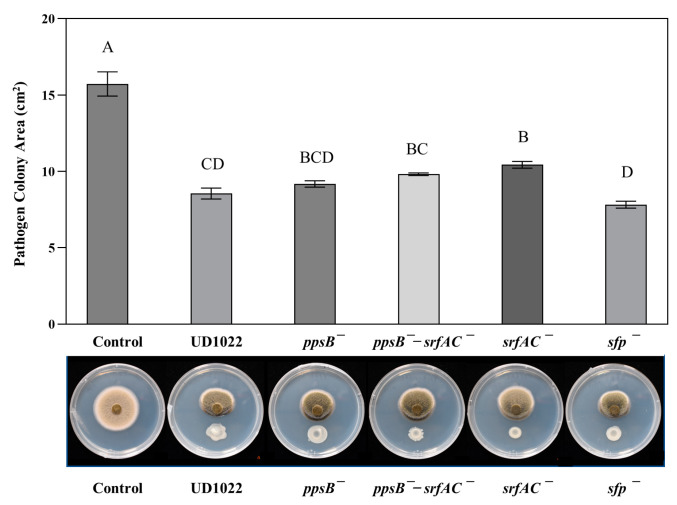
Results of antagonism assays with UD1022 mutants in key nonribosomal peptide (NRP) pathway genes against *A. medicaginicola* StC 306-5. Bars are the mean colony area (*n* = 3) and standard error (SE) of the mean from one representative experiment of three experiments performed. UD1022 mutant treatment *srfAC*^−^ was significantly less antagonistic than UD1022 (*p* = 0.044). The antagonistic effects of mutants *sfp*^−^, *ppsB*^−^, *ppsB*^−^-*srfAC*^−^ were no different from UD1022.

**Figure 3 plants-12-01007-f003:**
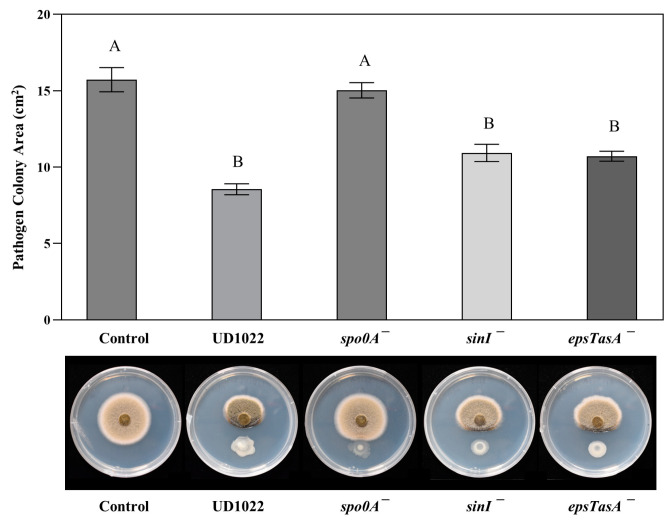
Results of antagonism assays with UD1022 mutants in key biofilm genes against *A. medicaginicola* StC 306-5. Bars are the mean colony area (*n* = 3) and standard error (SE) of the mean from one representative experiment of three experiments performed. The colony area of the *spo0A^−^* treatment was not significantly different from the control (*p* = 0.881), and biofilm mutant treatments sinI^−^ and *eps*^−^-*tasA*^−^ colony areas were not different from UD1022.

**Figure 4 plants-12-01007-f004:**
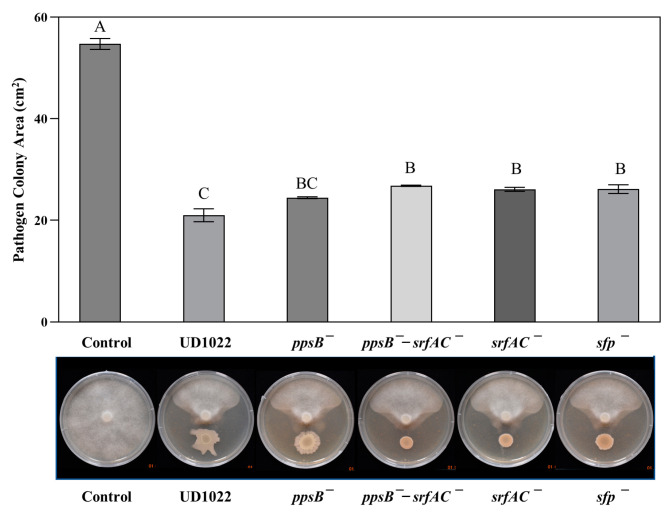
Results of antagonism assays with UD1022 mutants in key nonribosomal peptide (NRP) pathway genes against *P. medicaginis* A2A1. Bars are the mean colony area (*n* = 3) and standard error of the mean (SE) of the mean from one representative experiment of three experiments performed. The colony areas of the NRP mutant treatments, *ppsB*^−^-*srfAC*^−^, *srfAC*^−^, and *sfp*^−^ were significantly different from the UD1022 treatment (*p* = 0.002, 0.006, and 0.006, respectively). The *ppsB*^−^ treatment colony area was not significantly different from UD1022 (*p* = 0.073).

**Figure 5 plants-12-01007-f005:**
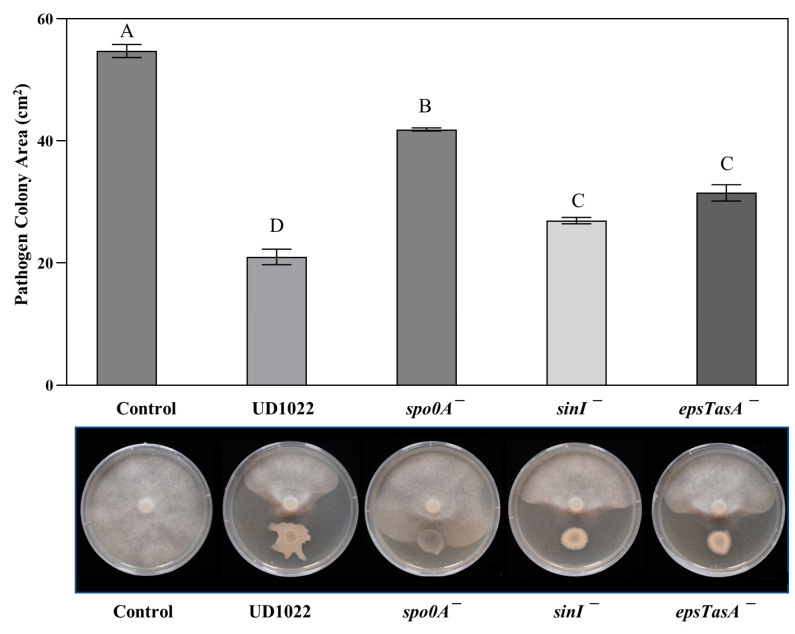
Results of antagonism assays with UD1022 mutants in key biofilm genes against *P. medicaginis* A2A1. Bars are the mean colony area (*n* = 3) and standard error of the mean (SE) of the mean from one representative experiment of three experiments performed. The *spo0A*^−^ treatment antagonism was significantly less than UD1022 (*p* < 0.0001), the *sinI*^−^ treatment was mildly less antagonized than UD1022 (*p* = 0.011), and the *eps*^−^-*tasA*^−^ double mutant was less antagonistic than UD1022 (*p* = 0.002).

## Data Availability

Data will be submitted in parallel to one of the data sharing sites.

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
