# Peer review of "Surfactin and Spo0A-Dependent Antagonism by Bacillus subtilis Strain UD1022 against Medicago sativa Phytopathogens"

_plants, 2023, doi:10.3390/plants12051007_

Round 1

Reviewer 1 Report

The authors evaluated the effects of a strain of Bacillus subtilis (UD1022) for its ability to antagonize M. sativa phytopathogens using in vitro tests and investigated possible modes of action, such as biofilm formation and secondary metabolites using UD1022 mutants.

The manuscript is well written, and I do believe the authors’ work is relevant and important in addition to fitting in the scope of your journal. I have reviewed this manuscript in a previous submission to another journal last year and I see now that the authors made the changes I had requested. I believe there are just a few minor changes/suggestions to be considered (below) and my recommendation is: accept with minor changes in Plants journal.

A few comments are listed below:

·      Abstract: add the goals of the work right after the intro sentence and put them in past tense.

·      Overall, the authors should consider writing the inhibition results in % to facilitate the reading and interpretation of the results in the abstract.

Author Response

Reviewer-I:

The authors evaluated the effects of a strain of Bacillus subtilis (UD1022) for its ability to antagonize M. sativa phytopathogens using in vitro tests and investigated possible modes of action, such as biofilm formation and secondary metabolites using UD1022 mutants.

The manuscript is well written, and I do believe the authors’ work is relevant and important in addition to fitting in the scope of your journal. I have reviewed this manuscript in a previous submission to another journal last year and I see now that the authors made the changes I had requested. I believe there are just a few minor changes/suggestions to be considered (below) and my recommendation is: accept with minor changes in Plants journal.

A few comments are listed below:

  • Abstract: add the goals of the work right after the intro sentence and put them in past tense.

The goal sentence has been revised to past tense and placed after the intro sentence.

  • Overall, the authors should consider writing the inhibition results in % to facilitate the reading and interpretation of the results in the abstract.

The authors have included the % inhibition for the treatments in the Results section.

Reviewer 2 Report

The article “Plant growth promoting rhizobacteria Bacillus subtilis strain 2 UD1022 antagonizes Medicago sativa phytopathogenes”  is scientifically very good. The presented results are interesting. The findings are promising, and further research is needed to investigate possible modes of action implicated in antagonism.

The introduction correctly places the study in a broad context and highlights its importance.

The description of the results is clear

The results are correctly discussed based on previous studies.

The methods used are sufficiently described.

The conclusion section is very long but I think it's appropriate and useful to highlight the limitations of the work.

However, I have some comments to be addressed:

The title does not precisely reflect the content of the text.

Line 55: How biofilm formation is considered as an indicator of antagonism in in-vitro bioassay conducted by the author!?

23-24: You must Specify a good candidate against which alfalfa disease!

Line 278: whether major characteristics of Bacillus spp.: here either you specify your strain or talk in general about the modes of action of biological control agents? I guest you mean your studied strain.

283 to 293 : Move to the introduction section

Line 305: Concerning the statistical analysis, the verification of the homogeneity of the variances and of the normality (ex. Shapiro–Wilk test) is not mentioned in the statistical analysis part, they must be mentioned if they are carried out since the choice of the type of statistics (parametric or nonparametric) depends on that.

Finally; The authors concentrated on the study of mutated strains and overlooked other modes of action which could have an important role in the antagonism. In my opinion, although the authors gave evidence of some mechanisms involved in this antagonism, they failed to study the mechanism of action in its entirety. It is true that the answer to this issue requires several works of the cost it is necessary to adapt the objective (to investigate possible modes of action for observed antagonisms) of the present studies on the basis of the results obtained.

Author Response

Reviewer-II

The article “Plant growth promoting rhizobacteria Bacillus subtilis strain 2 UD1022 antagonizes Medicago sativa phytopathogenes”  is scientifically very good. The presented results are interesting. The findings are promising, and further research is needed to investigate possible modes of action implicated in antagonism.

The introduction correctly places the study in a broad context and highlights its importance.

The description of the results is clear

The results are correctly discussed based on previous studies.

The methods used are sufficiently described.

The conclusion section is very long but I think it's appropriate and useful to highlight the limitations of the work.

However, I have some comments to be addressed:

The title does not precisely reflect the content of the text.

The authors have altered the title to describe the content of the article more specifically.

Line 55: How biofilm formation is considered as an indicator of antagonism in in-vitro bioassay conducted by the author!?

The authors are attempting to highlight the known importance of biofilm formation to PGP activities of Bacillus and that few works have investigated the possible direct or indirect mechanisms which biofilm may be contributing to pathogen inhibition (e.g., through superior root occupancy and association).

23-24: You must Specify a good candidate against which alfalfa disease!

The candidates have been specified by species of fungal pathogen at the conclusion of the abstract.

Line 278: whether major characteristics of Bacillus spp.: here either you specify your strain or talk in general about the modes of action of biological control agents? I guest you mean your studied strain.

We updated the sentence to specify that these characteristics of UD1022 are shared by B. subtilis and may be generalized across B. subtilis.

283 to 293 : Move to the introduction section

Section has been moved to the Introduction.

Line 305: Concerning the statistical analysis, the verification of the homogeneity of the variances and of the normality (ex. Shapiro–Wilk test) is not mentioned in the statistical analysis part, they must be mentioned if they are carried out since the choice of the type of statistics (parametric or nonparametric) depends on that.

Description of statistical analysis was expanded to include the tests for normality and homogeneity that were performed.

Finally; The authors concentrated on the study of mutated strains and overlooked other modes of action which could have an important role in the antagonism. In my opinion, although the authors gave evidence of some mechanisms involved in this antagonism, they failed to study the mechanism of action in its entirety. It is true that the answer to this issue requires several works of the cost it is necessary to adapt the objective (to investigate possible modes of action for observed antagonisms) of the present studies on the basis of the results obtained.

 Authors agree that further investigations into the mechanisms delineated in this work are needed and have added a short sentence acknowledging these shortcomings at the end of the conclusion, while maintaining the importance of the work in context of PGPR research for crop protection purposes.

Reviewer 3 Report

The present article “Plant growth promoting rhizobacteria Bacillus subtilis strain UD1022 antagonizes Medicago sativa phytopathogens” is an interesting article and based on good theme.

Author have performed here only Invitro analysis, which might be not effective in the invivo analysis. However, I have one  more quarries, what was the  concentration of  Bacillus subtilis during antagonistic assay. As the author only mentioned line -259 – “Twenty microliters”, because if the  higher concentration of antagonistic have been used then no any relevancy in the practical application

Author Response

Reviewer-III

The present article “Plant growth promoting rhizobacteria Bacillus subtilis strain UD1022 antagonizes Medicago sativa phytopathogens” is an interesting article and based on good theme.

Author have performed here only Invitro analysis, which might be not effective in the invivo analysis. However, I have one  more quarries, what was the  concentration of  Bacillus subtilis during antagonistic assay. As the author only mentioned line -259 – “Twenty microliters”, because if the  higher concentration of antagonistic have been used then no any relevancy in the practical application

The authors agree that in vitro activities may not translate to in vivo systems. Regarding the concentration of B. subtilis, the authors would note that the quantity of pathogen is also ‘higher’ than that found on a plant; the objective of the plate assay is to visually observe the activities/interactions of the pathogen and the PGPR. Also, applications of PGPR to plants/soil often target 108 to 109 cells/mL.

Round 2

Reviewer 2 Report

all the requested changes have been made